Review

# The Janus framework of the integrated stress response: from homeostasis to maladaptation

Dogus M Altintas[1] , Marina Cerqua[1], Paolo M Comoglio[1] , Cédric Chaveroux[2]

**Every cell must adapt to environmental changes. When nutrients decrease, oxygen levels fall, or protein synthesis outpaces resources, cells activate stress pathways to restore balance. Among these, the integrated stress response (ISR) stands out for its capability to integrate diverse stress signals into a unified translational output. By temporarily slowing global protein synthesis while maintaining the selective translation of stress-adaptive factors, the ISR saves energy, redirects metabolism, and promotes either recovery or, if challenges surpass repair capacity, cell death. In many chronic diseases—including cancer, metabolic, inflammatory, and fibrotic disorders—ISR activity persists. Is this persistence merely a prolonged defensive phase, or does it represent a rewired, self-sustaining state with its own control mechanisms actively reshaping cell fate and disease? We argue that chronic ISR cannot be defined by time alone, challenging the monolithic view. It signifies a qualitative shift in regulation—from rhythmic homeostasis to entrenched maladaptation. Understanding this Janus framework is essential for elucidating the origins of pathology and for guiding future fundamental and translational research.**

## The Integrated Stress Response: an Acute and Protective Programme

Cells are constantly challenged by environmental fluctuations that threaten their energy balance and survival. The integrated stress response (ISR) provides an elegantly simple solution: four stress-sensing serine/threonine kinases—PERK, GCN2, PKR, and HRI—each monitor distinct forms of stress, including endoplasmic reticulum perturbation, translational stress, viral infection, or haem deficiency/oxidative stress (Dever et al, 1992; Harding et al, 1999; Pakos-Zebrucka et al, 2016). Despite sensing different inputs, all converge on a single phosphorylation event at serine 51 of the eukaryotic translation initiation factor eIF2α. This modification reduces ternary-complex recycling and globally slows translation

initiation, allowing the cell to conserve resources and restore equilibrium (Wek et al, 2006).

Paradoxically, this global repression of translation enables the selective synthesis of transcripts that contain upstream open reading frames (uORFs) within their 5′ untranslated regions (5′UTR) (Harding et al, 2000). Chief among these is *ATF4*, which encodes a transcription factor that promotes amino acid synthesis and import, thereby restoring metabolic balance (Harding et al, 2003). One of ATF4's targets, *GADD34*, recruits the phosphatase PP1 to dephosphorylate eIF2α, terminating the response and reactivating translation (Novoa et al, 2001). Through this negative feedback loop, the ISR operates as a pulse: protective, self-limiting, and reversible (Fig 1) (Pakos-Zebrucka et al, 2016). When stress is too severe or irreparable, canonical ISR signalling contributes to cell death (Lin et al, 2009; Iurlaro & Muñoz-Pinedo, 2016; Pakos-Zebrucka et al, 2016; Costa-Mattioli & Walter, 2020). This terminal branch represents an alternative resolution outcome of acute ISR activation. In this perspective, we therefore refer to the acute, self-limiting configuration of the pathway—leading either to recovery or to terminal elimination by cell death—as the canonical ISR. Non-resolving, chronic regimes are discussed below.

## ISR in Diseases: when the Canonical Pathway Fragments

Recent quantitative work has described such canonical dynamics over hours to days in single cells exposed to defined pharmacological or viral stressors, where the sequence from ATF4 induction to CHOP activation and GADD34 feedback can be modelled in detail (Wijaya et al, 2021; Klein et al, 2022; Burgers et al, 2025). However, these paradigms do not capture the slow, fluctuating, and spatially heterogeneous stresses that shape human chronic diseases.

In chronic diseases—including fibrotic disorders (Korfei et al, 2008; Santos-Ribeiro et al, 2023), neurodegenerative conditions (Radford et al, 2015; Hu et al, 2022), metabolic diseases (Ryoo, 2024), and cancer (Lines et al, 2023)—ISR activation is almost ubiquitously observed. Yet, the pathway rarely appears in its complete canonical form (Costa-Mattioli & Walter, 2020). The textbook

[1]IFOM ETS-The AIRC Institute of Molecular Oncology, Milano, Italy    [2]LBTI UMR CNRS 5305, Faculty of Pharmacy, University of Lyon, Claude Bernard University, Lyon, France

Correspondence: dogus.altintas@ifom.eu

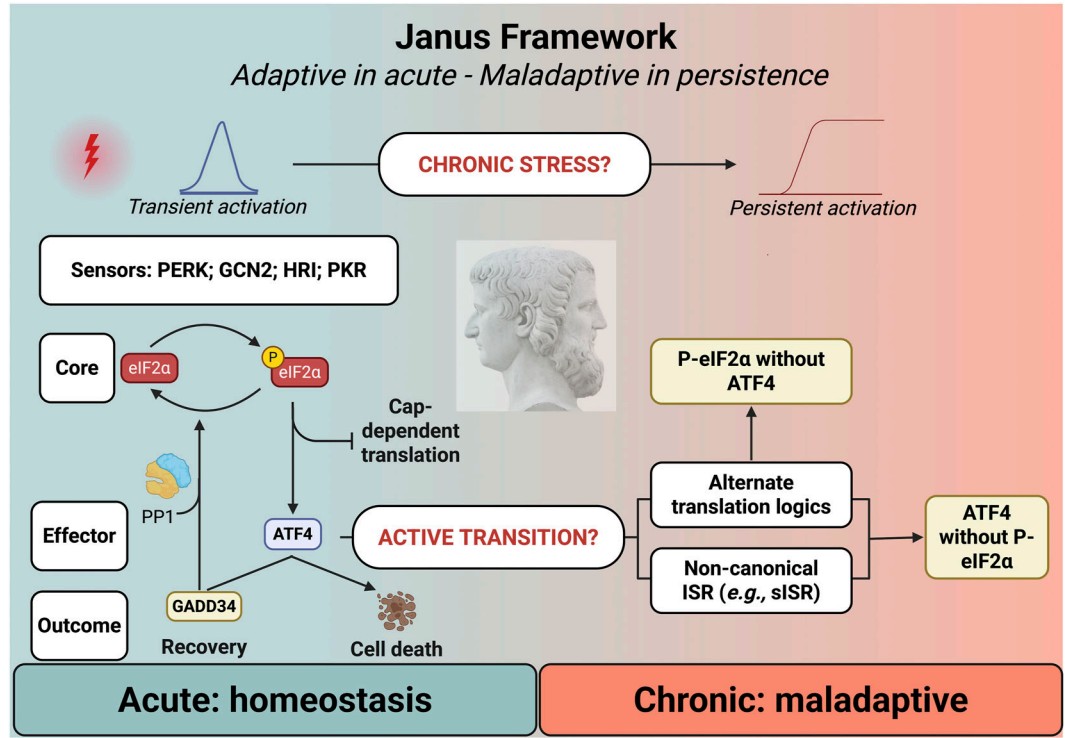

**Figure 1. Integrated stress response (ISR) acts as a Janus pathway—adaptive under acute stress, maladaptive when persistent.**
Acute ISR activation (canonical, blue) is transient and self-limiting through the ATF4-GADD34 feedback loop that restores homeostasis. Chronic ISR (red) may involve non-canonical routes (e.g., s-ISR) and alternative translational modes (e.g., eIF3-mediated re-initiation), producing decoupled outputs, such as eIF2α phosphorylation without ATF4, and sustained proteome rewiring that drives maladaptive states. This non-canonical state may emerge either actively, through distinct molecular circuitry, or through persistent stress as an adaptive response to a hostile environment. Understanding these regulatory regimes is crucial, as chronic ISR activation underlies key biological processes, including invasive growth. Created in https://BioRender.com.

sequence—sensor activation, eIF2α phosphorylation, ATF4 translational burst, GADD34 induction, and restoration of proteostasis—is seldom observed in its entirety. Instead, we observe a variety of mismatches that challenge the monolithic narrative of ISR.

One striking example comes from vanishing white matter (VWM) disease, caused by mutations in eIF2B, the guanine nucleotide exchange factor that recycles eIF2α between rounds of protein synthesis (Abbink et al, 2019). These mutations can lock cells into an ISR-like state even without elevated eIF2α phosphorylation. A recent study formally defined a "split ISR," in which reduced eIF2B activity engages a distinct ISR programme ("s-ISR") with altered translational control and gene expression, separable from the canonical eIF2α phosphorylation–dependent module. Functionally, the s-ISR rewires metabolism to maintain cellular bioenergetics and energy homeostasis when eIF2B activity is attenuated, for example, under mild hypoglycaemia (Chen et al, 2025). Similarly, in independent contexts, ATF4 was observed to accumulate in the apparent absence of eIF2α phosphorylation, for example, through mTORC1-dependent (Ben-Sahra et al, 2016) or eIF3-mediated re-initiation mechanisms (Guan et al, 2017).

Conversely, several studies have reported the opposite situation—persistent eIF2α phosphorylation without corresponding ATF4 accumulation (Kumar et al, 2003; Puri et al, 2008; Dey et al, 2010; Mesclon et al, 2017; Payea et al, 2024). Importantly, these non-canonical configurations are typically observed under chronic or unresolved stress in chronic diseases.

Together, these observations highlight that the canonical ISR is not sufficient to explain its diverse outputs, particularly in chronic diseases, underscoring the need to move beyond the monolithic view of the pathway (Fig 1).

## ISR in Disease: a Passive Consequence or an Active Disease–Defining State?

In chronic pathologies, cells thrive in a hostile microenvironment surrounded by toxic waste, apoptotic corpses, inflammatory cytokines, and fibrotic niches. Thus, it is not surprising that ISR activity is sustained in disease. However, the key question is whether this activation merely marks cellular distress or actively defines cellular trajectories through a wholesale reprogramming of the proteome.

In the lung, pharmacological attenuation of the ISR produces measurable therapeutic effects. The eIF2B activator ISRIB, which restores translation by counteracting eIF2α phosphorylation, ameliorates fibrosis and preserves epithelial differentiation in mouse models of idiopathic pulmonary fibrosis (Watanabe et al, 2021; Li et al, 2023). Such findings indicate that chronic ISR

engagement is not merely correlative but participates in shaping the fibrotic response.

Similarly, ATF4 signalling reprogrammes metabolism and drives sustained transcriptional reprogramming, reshaping cellular metabolism (Chaveroux et al, 2016), angiogenesis (Wang et al, 2013), and response to oxidative stress (Harding et al, 2003). This illustrates how the ISR can impose durable transcriptional and translational states, rewiring key biological functions well beyond the initial adaptive window (Chaveroux et al, 2016; Sarcinelli et al, 2020). Thus, persistent engagement does not remain neutral. Chronic ISR may stabilise long-lived transcriptional and translational programmes that actively contribute to disease by enabling cells to withstand hypoxia, nutrient limitation, or inflammatory signals (Bi et al, 2005; Lorenz et al, 2021; Cerqua et al, 2025). In these settings, ISR/ATF4-dependent gene expression programmes are required to maintain these maladaptive phenotypes, supporting the idea that chronic ISR can drive sustained transcriptional reprogramming. These maladaptive ISR-driven states may then be reinforced by cellular or clonal selection, which favours lineages in which ISR-dependent traits confer a survival, proliferative, or evolutionary advantage in the diseased microenvironment (Suh et al, 2014; Vendramin et al, 2021; Diao et al, 2025 *Preprint*).

Beyond the classical view as an on/off stress response pathway, its activity also governs cell fate trajectories (Laval et al, 2024). In muscle, satellite stem cells maintain low protein synthesis through eIF2$\alpha$ phosphorylation, a state essential for quiescence and self-renewal; blocking this phosphorylation forces premature activation, whereas sustaining it preserves regenerative capacity (Zismanov et al, 2016). In the epidermis, amino acid deprivation activates the ISR, which favours epidermal fate over hair follicle differentiation, thereby accelerating wound closure (Novak et al, 2025). In the haematopoietic system, ISR pathway activity—characterised by low eIF2$\alpha$ and high ATF4—marks primitive HSCs, promotes survival under nutrient stress, and distinguishes functional stem cells in both healthy and malignant conditions (van Galen et al, 2018).

Together, these findings reveal that the ISR is not merely a stress response but a conserved regulatory framework that dictates cellular trajectories across tissues and conditions. Its ability to reprogramme the proteome enables cells to persist in hostile environments—a principle prone to be hijacked in diseases, including cancer.

## From Adaptation to Invasion: ISR at the Crossroads of Stress and Oncogenic Signalling

Indeed, in cancer, ISR activation intersects with the invasive growth programme. Invasive growth is a physiological programme indispensable to survival under hostile environmental conditions and is central to a wide variety of biological processes. During embryogenesis, it governs morphogenetic events such as gastrulation and nervous system development; in adult tissues, it orchestrates inflammatory responses and wound healing through coordinated cell migration and extracellular matrix remodelling (Boccaccio &

Comoglio, 2006). This otherwise physiological programme is frequently usurped in cancer, where its pathological facet drives local invasion and distant metastasis, the ultimate stages of malignancy, responsible for ~90% of cancer-related deaths (Hanahan & Weinberg, 2011).

The master regulator of this programme, the *MET* proto-oncogene, stands out for its ability to coordinate all phases of invasive growth, from cytoskeletal rearrangements to matrix degradation and angiogenesis (Birchmeier et al, 2003; Comoglio et al, 2018). *MET* is an essential gene and remains singular among receptor tyrosine kinases in its ability to couple developmental survival with adult tissue resilience and repair (Bladt et al, 1995). It is pervasively overexpressed across human cancers (Altintas & Comoglio, 2023). Notably, this overexpression cannot be explained by genetic alterations or *MET* mRNA abundance, pointing to a layer of post-transcriptional control. Recent findings (Cerqua et al, 2025) reveal that this multifaceted programme is not only transcriptionally driven but also controlled through translational rewiring, providing a direct molecular link between the integrated stress response (ISR) and MET overexpression. In the hostile tumour microenvironment—characterised by hypoxia, nutrient scarcity, and chronic inflammation—tumour cells exploit the ISR-MET circuitry to exacerbate invasive growth. What begins as an adaptive response to stress thus becomes an engine of malignant progression and metastasis (Nguyen et al, 2018; Ghaddar et al, 2021), illustrating how chronic ISR activity can redefine oncogenic signalling with profound consequences for disease progression and therapy resistance.

## Outlook: Decoding the Janus Framework of the ISR

The ISR was long regarded as a linear and transient defence—activated when stress arises and silenced when equilibrium returns. Yet, evidence across systems now reveals a more profound paradox: the same pathway that preserves homeostasis under acute stress can, when chronically engaged, remodel cell identity and disease evolution. This duality defines the Janus framework of the ISR—its capacity to act both as a guardian of proteostasis and a driver of pathological plasticity (Fig 1) (Pakos-Zebrucka et al, 2016; Costa-Mattioli & Walter, 2020; Cerqua et al, 2025).

We argue that chronic ISR cannot be defined by time alone nor by the persistence of its molecular markers. Instead, it represents a qualitative shift in regulatory architecture, where feedback loops that typically reset translation become rewired or silenced, creating a new steady state (Guan et al, 2017; Chen et al, 2025). In this state, translation, metabolism, and cell fate are continuously reprogrammed to sustain survival in adverse conditions, possibly driving the maladaptive trajectories that characterise chronic diseases—from neurodegeneration (Radford et al, 2015) to fibrosis (Watanabe et al, 2021) and cancer (Ye et al, 2010; Cerqua et al, 2025). Recognising this shift transforms how we interpret ISR activity in disease: not as a prolonged alarm, but as a new operating mode for cell fate decisions, illustrating the Janus framework—a

translational hub defending the cell in crisis yet endowing pathological persistence and adaptability when locked on.

All these converging observations underscore the urgent need to decipher the molecular mechanisms underlying the chronic activation of ISR. This question lies at the intersection of fundamental biology, which helps us understand cellular adaptation processes, and translational biology, which reveals novel regulatory circuits essential for disease progression. However, addressing this challenge is easier said than done. Stress pathways are deeply intertwined; isolating the ISR from unfolded protein response (UPR) branches, mTOR, oxidative stress, and inflammatory signalling networks remains the central bottleneck. We therefore need to re-examine current models and build systems that stabilise and interrogate the chronic ISR state itself so that we can uncover its molecular features and its direct role in disease progression. This causal link remains only partially established. Pharmacologic attenuation (e.g., ISRIB in pulmonary fibrosis; GCN2 inhibitors in oncology) can improve phenotypes, but whether chronic ISR drives disease initiation/progression—or merely sustains it—remains unresolved.

To help address this research gap, we propose a set of working criteria—with built-in caveats—to distinguish acute, protective pulses from chronic, maladaptive states.

(1) Prioritise models that isolate and stabilise the chronic ISR state. Current tools were invaluable in defining the canonical ISR, but they also co-activated other stress programmes, blurring attribution. For instance, thapsigargin or tunicamycin robustly triggers PERK/ISR yet also engages broader ER-stress/UPR cascades; sodium arsenite activates HRI/oxidative stress alongside other damage pathways; poly(I:C) activates PKR and the interferon response. Phenotypes in these contexts cannot be confidently ascribed to the ISR framework alone and are poorly representative of physiological or disease-relevant states. A key step forward is the "split ISR" framework based on eIF2B genetic modulation in the absence of pharmacological perturbations (Chen et al, 2025), revealing an unsuspected new layer of ISR regulation, in line with challenging the monolithic view of the pathway.

(2) Loss of reversibility/hysteresis as a hallmark of chronic ISR? Acute ISR resolves when stress abates; in contrast, chronic ISR shows memory: persistence after stimulus withdrawal and/or sensitisation to a second hit (enhanced activation upon re-challenge). Although some authors proposed that this hysteretic behaviour likely reflects feedback fatigue or metabolic memory, the molecular basis of this persistence remains an open question, and whether this represents a context-dependent or general mechanism is still unknown (Radford et al, 2015; Costa-Mattioli & Walter, 2020).

(3) Translational divergence with distinct kinetics beyond the *ATF4* regulon. *ATF4* translation is typically transient, peaking early during stress and rapidly declining as feedback mechanisms restore homeostasis (Harding et al, 2000). In contrast, other ISR-responsive mRNAs—such as *MET*—show more sustained translation, supporting long-term adaptive programmes (Cerqua et al, 2025). Why these translational kinetics diverge remains unknown but experimentally testable, likely reflecting

differences in uORF architecture, transcript context, or interaction with auxiliary initiation factors.

Finally, acknowledging ISR's dual code—resilience versus pathology—should change how we read omics data. Because ISR reprogrammes translation, it can uncouple transcriptomes from proteomes, a discrepancy already documented in multiple disease contexts (Vogel & Marcotte, 2012; Liu et al, 2016). Before stratifying patients based solely on RNA proxies, we should integrate protein-level readouts of ISR activity to distinguish adaptive from maladaptive states correctly—and to more accurately link genotype to phenotype by incorporating this additional layer of translational control.

## Data Availability

This article is a conceptual perspective and does not report new data.

## Acknowledgements

DM Altintas was supported by the Fondazione Umberto Veronesi (Grant 3916). PM Comoglio was supported by the Fondazione AIRC per la Ricerca sul Cancro ETS (Grant 23820).

### Author Contributions

DM Altintas: conceptualisation, supervision, funding acquisition, visualisation, and writing—original draft, review, and editing.
M Cerqua: visualisation and writing—original draft.
PM Comoglio: conceptualisation, resources, supervision, and writing—original draft, review, and editing.
C Chaveroux: conceptualisation, supervision, and writing—original draft, review, and editing.

### Conflict of Interest Statement

The authors declare that they have no conflict of interest.

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
