## [Reviewer comments · Life Science Alliance]

The Janus Framework of the Integrated Stress Response: From Homeostasis to Maladaptation

Dogus Altintas, Marina Cerqua, Paolo Comoglio, and Cedric Chaveroux

DOI: <https://doi.org/10.26508/lsa.202503523>

Corresponding author(s): Dogus Altintas, IFOM

Review Timeline:

Submission Date:	2025-10-24
Editorial Decision:	2025-12-04
Revision Received:	2025-12-08
Editorial Decision:	2025-12-18
Revision Received:	2025-12-19
Accepted:	2025-12-19

Scientific Editor: Tim Fessenden

Transaction Report:

December 4, 2025

Re: Life Science Alliance manuscript #LSA-2025-03523-T

Dr. Dogus Murat Altintas
IFOM
IFOM
16 Via Adamello
Milano, Lombardy 20139
Italy

Dear Dr. Altintas,

Thank you for submitting your manuscript entitled "The Janus Logic of the Integrated Stress Response: From Homeostasis to Maladaptation" to Life Science Alliance. The manuscript was assessed by expert reviewers, whose comments are appended to this letter. We invite you to submit a revised manuscript addressing the Reviewer comments.

As you will see, reviewers overall appreciate this perspective and have provided a few suggestions to improve the clarity and accuracy of the text. We note that Reviewers 1 and 3 suggested replacing the "Janus" framework. Improving or replacing this framing device is left to your discretion.

While you are revising your manuscript, please also attend to the below editorial points to help expedite the publication of your manuscript. Please direct any editorial questions to the journal office. When submitting the revision, please include a letter addressing the reviewers' comments point by point.

Thank you for this interesting contribution to Life Science Alliance. We hope that the comments below will prove constructive as your work progresses, and we are looking forward to receiving your revised manuscript.

Sincerely,

B. MANUSCRIPT ORGANIZATION AND FORMATTING:

Reviewer #1 (Comments to the Authors (Required)):

This article by Altintas and colleagues describe in detail the role of the integrated stress response with a chosen angle of attack focusing on the dual aspect of this signaling pathway, adaptive on the one hand and terminal on the other.

Although the article provides a synthetic presentation of the ISR, it appears to be more in line with an opinion/perspective article that presents the authors reflexion on the nature of the ISR than a genuine review. First of all, the authors describe the acute and transient activation of the ISR, which operates when stress is resolved and how this mechanism is possibly highjacked upon chronic activation of the ISR. Second, the authors attempt to redefine the different aspects of the ISR

Major points

- The authors use the terminology « canonical ISR » but do not precisely define it, this should be fixed.
- The authors envision the ISR with a « janus » perspective, but in this scheme there is a missing arm, which is the terminal arm of the ISR that is not covered here. This aspect should be added to the manuscript, and therefore the « janus » presentation of the ISR should be abandoned, a three axes model could be presented. In addition, this could also help the author for pushing there views regarding the « reprogramming of the ISR beyond canonical » in a context of adaptation/death and survival upon chronic stress underlying an underlying selection process.
- The use of terms such as « logic » implies a deterministic view of the ISR which in my opinion is not accurate. A selection/evolutionary-based process would rather be more likely to exist.
- The balance between death and escape to canonical ISR should be further detailed, and could be also be put in perspective regarding the introduction of split ISR.

Minor points

- The figure should be revised according to the text
- Terms such as « logic » « janus » should be modified
- Defining passive consequence or active disease defining should be further documented as well, again in this perspective, the selection aspect should be introduced.

Reviewer #2 (Comments to the Authors (Required)):

The Review/Opinion piece from Altintas et al provides a discussion of conceptual advances in research on the Integrated Stress Resposne (ISR). The main thesis centers around consideration that the outputs of the ISR, which is known to try to maintain or reset homeostasis - and then, as the authors state nicely in the abstract - "if challenges surpass repair capacity, lead to cell death", are not simply coordinated through a timing-based mechanism. Rather, in chronic disease conditions in particular, long-term or persistent ISR represents an altered, maladapted steady state that contributes significantly to disease.

For the article, this thesis nicely holds together as the authors mention and briefly consider examples of non-linear ISR outputs and maladaptation. The recommendation from this reviewer to enhance the piece is to weigh the argument for the reader in a more balanced way how a timing-based mechanisms could also play a role in dictating different outputs. For example, what is the key evidence for a timing-based mechanism and in what context(s)? How would a timing mechanism work molecularly? Could timing normally underlie differential cell fates, while maladaptation could be driven in disease contexts by perturbations to this normal mechanism - genetic or otherwise? Some balance and consideration of this concept that is used as counterpoint would help to place the topic in better context for the reader, and could amplify the central thesis of an otherwise well-written and well-considered piece.

Reviewer #3 (Comments to the Authors (Required)):

In this brief, well-written perspective the authors summarize the functions of the Integrated Stress Response (ISR), a dynamic stress-responsive cascade that allows cells to react to environmental stressors to promote cell recovery or, if stress is

overwhelming, trigger cell death. The ISR relies on 4 serine/threonine kinases-PERK, GCN2, PKR and HRI-to detect and respond to unique stressors: ER stress, nutritional deprivation, viral infection and oxidative stress, respectively. Despite their distinct roles, these kinases all act through a single downstream kinase, eIF2 α , whose phosphorylation activates a signaling cascade to reprogram translation via ATF4. Traditionally, this reprogramming is thought to be transient and reversible. The authors briefly summarize evidence that this model is incomplete, particularly in chronic disease, in which translational reprogramming often proceeds in non-canonical fashion. They highlight how ISR activation exerts direct pathogenic effects in chronic disease, while also fulfilling essential roles in embryogenesis, lineage specification, and stem cell maintenance, and is further required for tumor invasion and metastasis by enabling cellular survival under adverse environmental conditions. Given the diverse consequences of the ISR, the authors argue, our traditional understanding of the ISR must be revised to account for this complexity. We must, they argue, seek to explain how the ISR both preserves homeostasis on one hand and contributes to disease on the other. They identify 3 priorities: (1) using genetic rather than pharmacologic models to isolate ISR physiology from other cell stress mechanisms, (2) interrogating the molecular basis by which chronic ISR predisposes to persistent transcriptional reprogramming and (3) deciphering how eIF2 α phosphorylation influences translational effects beyond those attributable to ATF4. While these suggestions certainly identify essential gaps in our understanding of the ISR, the authors' suggestions are not particularly innovative. Nonetheless, this is a strong manuscript.

Suggestions for Improvement:

1. A slightly more extensive discussion of the evidence that the ISR contributes to "sustained transcriptional reprogramming" since this a relatively underappreciated consequence of ISR activity (see lines 89-94) and is critical to their argument that the ISR causes a "qualitative shift" in the proteome. Furthermore, the adaptive/proapoptotic paradigm that the authors challenge has been reviewed in detail elsewhere and, thus, can be reviewed very briefly.
2. A minor point is the use of "Janus logic" to frame the argument. It is an unnecessary flourish that does not clarify their argument. Removing this framing would be straightforward and require minimal editing.

Point-by-point answers are written in bold characters below each comment.

Reviewer #1 (Comments to the Authors (Required)):

This article by Altintas and colleagues describe in detail the role of the integrated stress response with a chosen angle of attack focusing on the dual aspect of this signaling pathway, adaptive on the one hand and terminal on the other.

Although the article provides a synthetic presentation of the ISR, it appears to be more in line with an opinion/perspective article that presents the authors reflexion on the nature of the ISR than a genuine review. First of all, the authors describe the acute and transient activation of the ISR, which operates when stress is resolved and how this mechanism is possibly highjacked upon chronic activation of the ISR. Second, the authors attempt to redefine the different aspects of the ISR.

We thank the reviewer for this positive and thoughtful overview of our Perspective. We appreciate the recognition of the manuscript's conceptual angle and are grateful for the constructive suggestions that helped us clarify and strengthen the framework we propose.

Major points

- The authors use the terminology « canonical ISR » but do not precisely define it, this should be fixed.

We thank the reviewer for this helpful suggestion. We have now made explicit in the first section that it describes what we refer to as the canonical ISR. Specifically, we added a sentence at the end of this section stating that the acute, self-limiting and reversible ISR configuration—sensor activation, eIF2 α phosphorylation, ATF4 induction, GADD34 feedback and restoration of proteostasis—is what we term the canonical ISR (page 1, section 1). We use this definition consistently throughout the manuscript when contrasting canonical vs chronic, non-canonical ISR states.

- The authors envision the ISR with a « janus » perspective, but in this scheme there is a missing arm, which is the terminal arm of the ISR that is not covered here. This aspect should be added to the manuscript, and therefore the « janus » presentation of the ISR should be abandoned, a three axes model could be presented. In addition, this could also help the author for pushing there views regarding the « reprogramming of the ISR beyond canonical » in a context of adaptation/death and survival upon chronic stress underlying an underlying selection process.

We thank the reviewer for this thoughtful comment. We now clarify in the manuscript that the Janus metaphor is used to distinguish resolution of stress—whether through recovery or, when damage is irreparable, terminal elimination by cell death—from non-resolution, which corresponds to chronic

ISR engagement. In this sense, the metaphor reflects the distinction between canonical (acute, self-limiting) and non-canonical (persistent, maladaptive) ISR configurations rather than implying a strict two-state or deterministic model. We explicitly state in the revised text that resolution encompasses both homeostatic recovery and ATF4-dependent pro-apoptotic outputs leading to cell death (page 1, section 1).

In contrast, non-resolution denotes the chronic regime in which ISR signalling fails to terminate and instead stabilises maladaptive survival programmes—the main focus of this Perspective. This framing naturally accommodates the three outcome directions highlighted by the reviewer (recovery, death, maladaptive survival) while preserving the conceptual function of the Janus metaphor.

As apoptosis is not the focus of this Opinion piece and has already been extensively reviewed elsewhere, we briefly discuss the terminal arm to position it within the overall conceptual framework, while maintaining our emphasis on chronic, maladaptive ISR states.

- The use of terms such as « logic » implies a deterministic view of the ISR which in my opinion is not accurate. A selection/evolutionary-based process would rather be more likely to exist.

We thank the reviewer for this critical comment and we agree entirely with the concern raised. To avoid any deterministic implication, we have replaced the term “logic” throughout the manuscript, including in the title, which is now:

“The Janus Framework of the Integrated Stress Response: From Homeostasis to Maladaptation.”

We chose the term “framework” precisely because it does not imply a fixed or mechanistic decision tree. Instead, it denotes a flexible conceptual structure that accommodates probabilistic cell-fate tendencies, context-dependent outputs, and the evolutionary/selection-based filtering processes.

- The balance between death and escape to canonical ISR should be further detailed, and could be also be put in perspective regarding the introduction of split ISR.

We thank the reviewer for this helpful suggestion. In the revised manuscript, we clarify that canonical ISR can resolve either through homeostatic recovery or, when damage is irreparable, through ATF4-dependent terminal cell-death programmes, and we explicitly define both outcomes as part of the acute canonical ISR (page 1, paragraph 1). In addition, when introducing the concept of split ISR, we now explain that reduced eIF2B activity engages a distinct ISR programme (‘s-ISR’) with altered translational control and metabolic rewiring that allows cells to deviate from the canonical ISR trajectory that normally terminates in recovery or death (page 2, paragraph 2; Chen C-W et al, 2025). This, we believe, places the balance between death and escape from canonical ISR into the broader context of non-canonical ISR regimes.

Minor points

- The figure should be revised according to the text

We thank the reviewer. The figure has been revised to match the updated conceptual framework presented in the text.

- Terms such as « logic » « janus » should be modified

We agree. We have replaced “logic” throughout the manuscript and updated the title to “The Janus Framework of the Integrated Stress Response: From Homeostasis to Maladaptation.” The revised terminology avoids deterministic connotations while preserving the intended conceptual framing.

- Defining passive consequence or active disease defining should be further documented as well, again in this perspective, the selection aspect should be introduced.

We thank the reviewer for this helpful suggestion. We have now clarified the distinction between ISR activation as a passive consequence of tissue stress and chronic ISR as an active disease-defining driver. In the revised text (page 3, paragraph 3), we added a sentence noting that maladaptive ISR states can confer selective advantages under adverse microenvironmental conditions and can therefore be reinforced by cellular or clonal selection.

Reviewer #2 (Comments to the Authors (Required)):

The Review/Opinion piece from Altintas et al provides a discussion of conceptual advances in research on the Integrated Stress Response (ISR). The main thesis centers around consideration that the outputs of the ISR, which is known to try to maintain or reset homeostasis - and then, as the authors state nicely in the abstract - "if challenges surpass repair capacity, lead to cell death", are not simply coordinated through a timing-based mechanism. Rather, in chronic disease conditions in particular, long-term or persistent ISR represents an altered, maladapted steady state that contributes significantly to disease.

For the article, this thesis nicely holds together as the authors mention and briefly consider examples of non-linear ISR outputs and maladaptation. The recommendation from this reviewer to enhance the piece is to weigh the argument for the reader in a more balanced way how a timing-based mechanisms could also play a role in dictating different outputs. For example, what is the key evidence for a timing-based mechanism and in what context(s)? How would a timing mechanism work molecularly? Could timing normally underlie differential cell fates, while maladaptation could be driven in disease contexts by perturbations to this normal mechanism - genetic or otherwise? Some balance and consideration of this concept that is used as counterpoint would help to place the topic in better context for the reader, and could amplify the central thesis of an otherwise well-written and well-considered piece.

We thank the reviewer for these helpful and thoughtful comments. We agree that timing-based mechanisms have played an important role in classical interpretations of acute ISR dynamics. In the revised manuscript, we explicitly acknowledge this work by referring to studies in which temporal features of the ATF4–CHOP axis, together with delayed GADD34 feedback, have been quantitatively modelled over hours to days in single cells, and shown to predict cell death or to provide short-term “molecular memory” of prior stress (Wijaya et al, 2021; Burgers et al, 2025; Klein et al, 2022) (page 1, section 1).

We then clarify that these studies primarily address acute and subacute ISR dynamics under well-defined stress, and that it remains unknown how such timing-based frameworks extend to the much longer timescales and fluctuating, spatially heterogeneous stresses that characterise human chronic diseases. We hope this revision makes the distinction between timing-based and state-based views of the ISR clearer.

Reviewer #3 (Comments to the Authors (Required)):

In this brief, well-written perspective the authors summarize the functions of the Integrated Stress Response (ISR), a dynamic stress-responsive cascade that allows cells to react to environmental stressors to promote cell recovery or, if stress is overwhelming, trigger cell death. The ISR relies on 4 serine/threonine kinases-PERK, GCN2, PKR and HRI-to detect and respond to unique stressors: ER stress, nutritional deprivation, viral infection and oxidative stress, respectively. Despite their distinct roles, these kinases all act through a single downstream kinase, eIF2 α , whose phosphorylation activates a signaling cascade to reprogram translation via ATF4. Traditionally, this reprogramming is thought to be transient and reversible. The authors briefly summarize evidence that this model is incomplete, particularly in chronic disease, in which translational reprogramming often proceeds in non-canonical fashion. They highlight how ISR activation exerts direct pathogenic effects in chronic disease, while also fulfilling essential roles in embryogenesis, lineage specification, and stem cell maintenance, and is further required for tumor invasion and metastasis by enabling cellular survival under adverse environmental conditions. Given the diverse consequences of the ISR, the authors argue, our traditional understanding of the ISR must be revised to account for this complexity. We must, they argue, seek to explain how the ISR both preserves homeostasis on one hand and contributes to disease on the other. They identify 3 priorities: (1) using genetic rather than pharmacologic models to isolate ISR physiology from other cell stress mechanisms, (2) interrogating the molecular basis by which chronic ISR predisposes to persistent transcriptional reprogramming and (3) deciphering how eIF2 α phosphorylation influences translational effects beyond those attributable to ATF4. While these suggestions certainly identify essential gaps in our understanding of the ISR, the authors' suggestions are not particularly innovative. Nonetheless, this is a strong manuscript.

We thank the reviewer for the positive evaluation and for clearly articulating how the Perspective fits into the current ISR literature. We appreciate the constructive suggestions, which have helped us refine the scope and clarify several key concepts in the revised manuscript.

Suggestions for Improvement:

1. A slightly more extensive discussion of the evidence that the ISR contributes to "sustained transcriptional reprogramming" since this a relatively underappreciated consequence of ISR activity (see lines 89-94) and is critical to their argument that the ISR causes a "qualitative shift" in the proteome. Furthermore, the adaptive/proapoptotic paradigm that the authors challenge has been reviewed in detail elsewhere and, thus, can be reviewed very briefly.

We thank the reviewer for raising this important point. Whether the ISR can durably reconfigure the transcriptome in a manner that persists beyond its initial activation remains a critical and open question. Precisely for this

reason, our Perspective emphasises the need to distinguish canonical (acute, reversible) ISR from chronic, non-resolving regimes, and we propose experimental frameworks to determine whether chronic ISR states exhibit hysteresis, loss of reversibility, or transcriptional “memory.” In line with the reviewer’s comment, we highlight this knowledge gap explicitly in the Outlook section (points 1–3), where we argue that addressing these questions requires refined ISR models and stress paradigms capable of isolating chronic ISR from confounding pathways or general toxicity of common reagents used to activate ISR. We now explicitly state that in several chronic disease models, ISR/ATF4-dependent gene-expression programmes may be required to maintain maladaptive phenotypes (page 3).

We agree with the reviewer that the adaptive/pro-apoptotic paradigm has been extensively reviewed and should be kept brief. In the revised manuscript, we summarise this classical view in a concise manner (page 1)—highlighting that acute ISR promotes recovery, whereas overwhelming stress can engage ATF4-dependent cell-death programmes—before shifting the focus to chronic, non-resolving ISR states, which are the main topic of this Perspective.

2. A minor point is the use of "Janus logic" to frame the argument. It is an unnecessary flourish that does not clarify their argument. Removing this framing would be straightforward and require minimal editing.

We thank the reviewer for this helpful remark. In response, we have softened the wording in the title and throughout the text, and we now briefly clarify in the text that this metaphor is used purely as a conceptual scaffold to distinguish resolution (recovery or cell death) from non-resolution (chronic maladaptive ISR), rather than as a mechanistic model. We hope this revision preserves the intended conceptual emphasis while avoiding any impression of unnecessary flourish.

December 18, 2025

RE: Life Science Alliance Manuscript #LSA-2025-03523-TR

Dr. Dogus Murat Altintas
IFOM - FIRC Institute of Molecular Oncology
16 Via Adamello
Milano, Lombardy 20139
ITALY

Dear Dr. Altintas,

Thank you for submitting your revised review entitled "The Janus Framework of the Integrated Stress Response: From Homeostasis to Maladaptation". We would be happy to publish your paper in Life Science Alliance pending any final revisions outlined below.

Please upload the following materials to our submission system by logging in to your account: <https://lsa.msubmit.net/cgi-bin/main.plex>

You will be guided to complete the submission of your revised manuscript.

-- High-resolution figure, supplementary figure and video files uploaded as individual files: See our detailed guidelines for preparing your production-ready images, <https://life-science-alliance.org/authorguide>

-- Please add the X and Bluesky handles of your host institute/organization, as well as your own and/or one of the authors, in our system.

The license to publish form must be signed before your manuscript can be sent to production. A link to the license to publish form will be sent to the corresponding author only. Please take a moment to check your funder requirements.

Thank you for this interesting contribution, we look forward to publishing your Research Article in Life Science Alliance.

Sincerely,

Reviewer #1 (Comments to the Authors (Required)):

The authors properly addressed my comments raised on the initial version of the manuscript, thereby providing a much improved revised version.

Reviewer #2 (Comments to the Authors (Required)):

The review has been improved significantly by the changes that were made, this reviewer has no further comments.

Congratulations on a nice piece of work.

December 19, 2025

RE: Life Science Alliance Manuscript #LSA-2025-03523-TRR

Dr. Dogus Murat Altintas
IFOM
16 Via Adamello
Milano, Lombardy 20139
Italy

Dear Dr. Altintas,

Thank you for submitting your Review article entitled "The Janus Framework of the Integrated Stress Response: From Homeostasis to Maladaptation". It is a pleasure to let you know that your manuscript is now accepted for publication in Life Science Alliance. Congratulations on this interesting work and thank you for bringing your intriguing perspective on this topic to LSA.

Your manuscript will now progress through copyediting and proofing. It is journal policy that authors provide original data upon request. Please note LSA does not have a Perspective manuscript type and this will be published as a Commissioned Review.

DISTRIBUTION OF MATERIALS:

Again, congratulations on a very nice paper. I hope you found the review process to be constructive and are pleased with how the manuscript was handled editorially. We look forward to future exciting submissions from your lab.

Sincerely,
